



# A comprehensive organic nitrate chemistry: insights into the lifetime of atmospheric organic nitrates

Azimeh Zare[1], Paul S. Romer[1], Tran Nguyen[2], Frank N. Keutsch[3,a], Kate Skog[3,b], Ronald C. Cohen[1, 4]

[1]Department of Chemistry, University of California Berkeley, Berkeley, CA, USA
[2]College of Agricultural and Environmental Sciences, University of California, Davis, CA, USA
[3]Department of Chemistry, University of Wisconsin-Madison, Madison, WI, USA
[a]now at: School of Engineering and Applied Sciences and Department of Chemistry & Chemical Biology, Harvard University, Cambridge, MA, USA
[b]now at: Department of Chemical & Environmental Engineering, Yale University, New Haven, CT, USA
[4]Department of Earth and Planetary Sciences, University of California Berkeley, Berkeley, CA, USA

*Correspondence to*: Ronald C. Cohen (rccohen@berkeley.edu)

**Abstract.** Organic nitrate chemistry is the primary control over the lifetime of nitrogen oxides (NOx) in rural and remote continental locations. As NOx emissions decrease, organic nitrate chemistry becomes increasingly important to urban air quality. However, the lifetime of individual organic nitrates and the reactions that lead to their production and removal
remain relatively poorly constrained, causing organic nitrates to be poorly represented by models. Guided by recent laboratory and field studies, we developed a detailed gas phase chemical mechanism representing most of the important individual organic nitrates. We use this mechanism within the WRF-Chem model to describe the role of organic nitrates in nitrogen oxide chemistry and in comparisons to observations. We find the daytime lifetime of total organic nitrates with respect to all loss mechanisms to be 2.6 h in the model. This is consistent with analyses of observations at a rural site in
central Alabama during the Southern Oxidant and Aerosol Study (SOAS) in summer 2013. The lifetime of the first-generation organic nitrates is ~2 h versus the 3.2 h lifetime of secondary nitrates produced by oxidation of the first-generation nitrates. The different generations are subject to different losses, with dry deposition to the surface dominant loss process for the second-generation organic nitrates, and chemical loss dominant for the first-generation organic nitrates. Removal by hydrolysis is found to be responsible for the loss of ~30% of the total organic nitrate pool.

**1 Introduction**

In remote continental regions, biogenic volatile organic compounds (BVOCs), including isoprene and terpenes, are the most reactive organic compounds in the atmosphere (Guenther, 2013). The oxidative chemistry of BVOCs affects the distribution of oxidants (OH, $O_3$, $NO_3$) and the lifetime of $NO_x$ (=NO+$NO_2$), creating a feedback loop that affects oxidant concentrations, the lifetime of BVOCs and secondary organic aerosol formation. Along the pathway to complete oxidation of BVOCs,
reactions with the nitrogen oxide family radicals (NO, $NO_2$ and $NO_3$) to form organic nitrate products (e.g. Perring et al., 2013) are an important branch point that sets the importance of this feedback.



During the day, BVOCs react with the hydroxyl radical (HO) and peroxy radicals ($RO_2$) are formed. At modest concentrations of $NO_x$, the peroxy radicals react primarily with NO. The major products of that reaction are $NO_2$ and an alkoxy radical (RO). There is also a minor channel (with a branching fraction α) that results in addition of the NO to the peroxy radical resulting in an organic nitrate ($RONO_2$) product. During the night, nitrate radicals ($NO_3$), the product of the

oxidation of $NO_2$ by $O_3$, are also a major source of $RONO_2$. BVOCs react with $NO_3$, resulting in the formation of nitrooxy peroxy radicals in high yields. The radicals subsequently react to form closed shell $RONO_2$, with branching ratio β.

In the last decade, there have been major updates to our understanding of the chemical reactions that occur during isoprene oxidation (Paulot et al., 2009a, 2009b; Crounse et al., 2011; Liu et al., 2013; Peeters et al., 2014; Nguyen et al., 2014; Wolfe et al., 2016; Mills et al., 2016; Teng et al., 2017). This understanding includes recognition that the yield of $RONO_2$ from

reaction of isoprene peroxy radicals with NO is 11–15%, at the high end of the range reported in earlier laboratory experiments (Wennberg et al., 2018) The yield of nitrates from monoterpene oxidation is less clear as laboratory data indicate a very wide range (e.g. from greater than 1% (Aschmann et al., 2002) to 26% (Rindelaub et al., 2015)). For $NO_3$ oxidation of isoprene experimental data show that the yield, β, is high and varies in the range of 65%−80% (Perring et al., 2009a; Rollins et al., 2009; Kwan et al., 2012).

Once formed, $RONO_2$ can be photolyzed or oxidized to produce $NO_x$ or $HNO_3$ along with an organic partner, or they can serve as reservoirs of $NO_x$ that can be transported or deposited to the surface. An additional pathway for gas-phase $RONO_2$ loss is partitioning into aerosol in either an organic phase where vapor pressure would describe partitioning or an aqueous phase where a Henry's law constant would describe solubility. In the aerosol, the $RONO_2$ can undergo liquid phase reaction. Some $RONO_2$ are rapidly hydrolyzed with time scales on the order of hours to minutes under environmentally-relevant pH

conditions (Jacobs et al., 2014; Boyd et al., 2015; Rindelaub et al., 2016), while other nitrates are thought to be relatively stable against hydrolysis in neutral conditions (Hu et al., 2011). The main nitrogen-containing product of organic nitrate hydrolysis is nitric acid (Darer et al., 2011). Using measurements of organic nitrates and nitric acid over the Canadian boreal forest and southeast US, Browne et al. (2013) and Romer et al. (2016) provide evidence that hydrolysis of monoterpene and isoprene nitrates is likely a significant loss process and contributes to $HNO_3$ production. The short lifetime of $HNO_3$ to

deposition in the boundary layer means that organic nitrate loss through hydrolysis in the boundary layer is a permanent sink of $NO_x$.

For any organic nitrate, its structure determines the rate of its oxidation and photolysis as well as the rate of hydrolysis and deposition. Multifunctional nitrates containing hydroxyl or peroxide groups are likely to have deposition rates much faster than the rates for monofunctional nitrates (Shepson et al., 1996). The dry deposition of organic nitrates has been discussed in

the studies by Farmer and Cohen (2008) and Nguyen et al. (2015). Nguyen et al. (2015) directly measured deposition rates of organic nitrates from BVOCs and the first-generation isoprene nitrates were observed to have daytime dry deposition velocity of ~ 2 cm s$^{-1}$, which is higher than the values currently used in most models (Ito et al., 2009; Mao et al., 2013; Browne et al., 2014).



Unlike hydrolysis of organic nitrates in aerosol and deposition of organic nitrates to the surface, which is considered a sink of nitrogen oxides in the atmosphere, oxidation and photolysis of $RONO_2$ may recycle $NO_x$. Different assumptions regarding $NO_x$ recycling during organic nitrate oxidation result in large variations in simulation of $NO_x$ and $O_3$ (von Kuhlmann et al., 2004; Fiore et al., 2005; Wu et al., 2007; Horowitz et al., 2007; Paulot et al., 2012). For example, Xie et al. (2013) showed

that the uncertainty in the fraction of $NO_x$ returned to the atmosphere during isoprene nitrate oxidation had a larger impact than uncertainty in isoprene nitrate yield on $O_3$ production. This affirms the need for characterization of the fate and lifetime of $RONO_2$ in the atmosphere. New clarity is available for the chemical fate of the first-generation isoprene nitrates (e.g. (Lee et al., 2014; Xiong et al., 2015, 2016), while much less is known about the fate of organic nitrates formed from monoterpenes. Because few of these loss processes have been measured, especially for highly oxidized or monoterpene

nitrates, there is large uncertainty associated with any description of the lifetime of organic nitrates. Several modeling studies (Paulot et al., 2012; Xie et al., 2013; Mao et al., 2013) have focused specifically on the fate of isoprene nitrates, and have found that how their chemistry is represented has major consequences for $NO_x$ and $O_3$. Recently, Browne et al., (2014) extended the representation of organic nitrate chemistry by including in detail the gas-phase chemistry of monoterpenes and discussed different scenarios for uncertain loss processes of monoterpene nitrates. Their improved mechanism for BVOC

chemistry has been used as a skeleton for several subsequent modeling studies (e.g. Fisher et al., 2016 and this work). However, none of these models has yet combined detailed molecular representations of individual $RONO_2$ derived from anthropogenic, isoprene, and monoterpene VOC precursors. Here we describe the development of a gas phase mechanism along those lines. In a forthcoming paper we couple the mechanism described here to aerosol and cloud properties. Here we approximate the effects of aerosols and clouds with simpler parameters representing the effects of the condensed phase

chemistry. The model calculations are compared to observations from the SOAS (the Southern Oxidant and Aerosol Study) campaign in the rural Southeastern United States in summer 2013. We explore the relative contributions of OH and $NO_3$ chemistry to the production of organic nitrates from BVOCs and investigate the importance of different organic nitrate loss processes. Then we explore the lifetime of organic nitrates and consequences of organic nitrate chemistry for atmospheric $NO_x$ to understand the role of $RONO_2$ in the $NO_x$ and ozone budgets in the moderate $NO_x$, BVOC-dominated terrestrial

environments that represent the most common chemical regime on the continents during summer.

## 2 Model Description; WRF-Chem model

We use WRF-Chem version 3.5.1 (Grell et al., 2005) with a horizontal resolution of 12 km and 30 vertical layers over the eastern United States. Our simulation domain is defined on the Lambert projection, which is centered at 35°N, 87°W and has 290 and 200 grid points in the west–east and south–north directions, respectively (see Fig. 3 for the horizontal domain).

Meteorological data for initial and boundary conditions are driven by the North American Regional Reanalysis (NARR) and data for the chemical initial and boundary conditions are taken from MOZART (Emmons et al., 2010). The model simulation



period is from 27 May to 30 June 2013, with the first 5 days as spin-up, similar to Browne et al. (2014), to remove the impact of initial condition.

Anthropogenic emissions are based on the US EPA 2011 National Emission Inventory (NEI) and scaled to 2013 based on the changes in the annual average emissions from 2011 to 2013. The appropriate scale factors have been derived from the

NEI Air Pollutant Emissions Trend Data. We also adjust $NO_x$ emissions (uniformly reduced by 50%) following Travis et al. (2016), who suggest that reduced $NO_x$ emissions can better reproduce the SEAC4RS aircraft measurements for the Southeastern United States. Lightning emissions of $NO_x$ are not included in the model. Lightning $NO_x$ is mainly released at the top of convective updrafts (Ott et al., 2010) and does not strongly impact the distribution of $NO_2$ in the boundary layer (e.g. Laughner and Cohen, 2017). Biogenic emissions of isoprene, monoterpenes, other BVOCs, oxygenated VOCs

(OVOCs), and nitrogen gas emissions from the soil are parameterized using the Model of Emissions of Gases and Aerosol from Nature (MEGAN) (Guenther et al., 2006).

Gas phase reactions are simulated using the second generation Regional Atmospheric Chemistry Mechanism (RACM2) (Goliff et al., 2013), as updated by Browne et al. (2014) and with additions to the mechanisms as described below, which is implemented with the kinetic preprocessor (KPP) software (Damian et al., 2002). The Modal Aerosol Dynamics Model for

Europe/Secondary Organic Aerosol Model (MADE/SORGAM) aerosol module (Ackermann et al., 1998; Schell et al., 2001) is used to treat organic and inorganic aerosols. This two-product aerosol scheme does not treat the partitioning of individual chemical species such as organic nitrates. Therefore, we focus here only on investigating the impacts of the gas-phase representation of the chemistry and a full consideration of the gas and aerosol in a coupled framework is a subject of continuing research.

**2.1 Chemical Mechanism**

We base our core chemistry on a modified version of RACM2 (Goliff et al., 2013). The base we begin with is described by Browne and Cohen (2012) and Browne et al. (2014), and is referred to here as the RACM2_Berkeley scheme. Full details of the RACM2_Berkeley mechanism and a complete list of the compounds can be found in Browne et al. (2014). RACM2_Berkeley includes updates to the isoprene oxidation mechanism (Paulot et al., 2009a, 2009b; Peeters and Müller,

2010; Lockwood et al., 2010; Stavrakou et al., 2010; Crounse et al., 2011), an extended mechanism for anthropogenic-originated organic nitrates (Carter and Atkinson, 1989; Middleton et al., 1990; Arey et al., 2001) and updates for monoterpene chemistry. Browne et al. (2014) evaluated the RACM2_Berkeley mechanism using aircraft observations over the Canadian boreal forest.

In this study, the RACM2_Berkeley scheme is further updated with recent advances in the representation of OH- and $NO_3$-

initiated BVOC oxidation under both low- and high-$NO_x$ conditions, as well as with improved deposition rates and is denoted RACM2_Berkeley2 (see Table S1-S3 in the Supplement). We begin with a more complete description of recent advances in our understanding of isoprene chemistry (Fig. 1). The hydroxy peroxy radical (ISOPO2) that is the product of




isoprene oxidation by OH has multiple potential fates. ISOPO2 can undergo unimolecular isomerization, leading to the production of hydroperoxy aldehydes (HPALD), among other products. It can react with HO2 to produce isoprene hydroxy hydroperoxide (ISHP), methyl vinyl ketone (MVK), methacrolein (MACR) and $CH_2O$. The latter three species can also be formed from the reactions of ISOPO2 with other (acetyl or methyl) peroxy radicals. ISHP react with OH to form isoprene

epoxydiols (IEPOX) and regenerate OH. St. Clair et al. (2015) found that the reaction rate of ISHP + OH is approximately 10% faster than the rate given by Paulot et al. (2009b) and indicate the relative role of the different isomers of ISHP. Here we use kinetics and products of the reactions of three different isomers of ISHP with OH based on St. Clair et al. (2015). We also increase the molar yield of total ISHP from the ISOPO2 + HO2 reaction to 93.7% (Liu et al., 2013), with a decrease in the yields of MVK, MACR and $HO_x$ to maintain mass balance. We use rates from Bates et al. (2016) for reactions of three

different isomers of IEPOX with OH.

We maintain the overall branching ratio of isoprene nitrates at 11.7% as in Browne et al. (2014) while changing the mix of isomers. Browne et al. (2014) implemented a scheme for the reaction of ISOPO2 with NO, based on experiments conducted by Paulot et al. (2009b), including β and δ-hydroxy isoprene nitrates (ISOPNB and ISOPND) with yields 4.7% and 7.0%, respectively. Here we update the yield of β versus δ isomers to 10.5% and 1.2% respectively. A theoretical study by Peeters

et al. (2014) showed that the peroxy radical redissociations are fast and peroxy isomers may interconvert, so that β-isomers comprise ∼95% of the radical pool. The experimental findings of Teng et al. (2017) are also consistent with that idea. Simulations by Fisher et al. (2016) showed that an isoprene hydroxy radical distribution leading to 90% ISOPNB and 10% ISOPND is consistent with SOAS observations.

ISOPND and ISOPNB then photolyze or react with O3 or OH to yield products that are either $NO_x$ or second-generation

organic nitrates. We update reaction rates of isoprene hydroxy nitrate oxidation based on Lee et al. (2014). Compared to the RACM2_Berkeley mechanism (based on Paulot et al. (2009b) and Lockwood et al. (2010)), the reaction rates for ISOPNs+OH are increased and the rate coefficients of ISOPNs+O3 are decreased. The model represents the products of the reactions of ISOPND and ISOPNB with OH as ISOPNDO2 and ISOPNBO2. We update the reaction of ISOPNB+OH to include a small yield of IEPOX and $NO_2$ (12%) as found by Jacobs et al. (2014). We also update the rate constants for

reaction of ISOPNDO2 and ISOPNBO2 with NO producing second-generation isoprene nitrates following Lee et al. (2014). Second-generation isoprene nitrates from the OH-initiated pathway include ethanal nitrate (ETHLN), propanone nitrate (PROPNN), multifunctional isoprene nitrate (IMONIT), methacrolein nitrate (MACRN) and methylvinylketone nitrate (MVKN). MACRN and MVKN can also be formed directly from photooxidation of MVK and MACR under high-$NO_x$ conditions. Here, we follow Praske et al. (2015) to update MVK chemistry under both low- and high-$NO_x$ conditions,

resulting in greater recycling of OH and $NO_2$ and decreased formation of organic nitrates from reactions with NO or HO2.

In addition to OH chemistry, isoprene is oxidized by NO3. In RACM2_Berkeley, the isoprene + NO3 chemistry was parameterized with one generic organic nitrate as the product. Recently, Schwantes et al. (2015) developed a kinetic



mechanism for $NO_3$-initiated oxidation of isoprene in which products, branching ratios, and rate constants are estimated based on recent experimental results. Their suggested products from $NO_3$ oxidation of isoprene are consistent with organic nitrates detected in the ambient atmosphere during SOAS. RACM2_Berkeley2 treats the $NO_3$ initiated oxidation of isoprene in some detail with formation and subsequent oxidation of isoprene nitrates largely based on the work of Schwantes et al.

(2015) and Rollins et al. (2009) (Fig. 2). In the first step, $NO_3$ addition to isoprene forms a nitrooxy peroxy radical ($INO_2$) and then, depending on the radical the $INO_2$ reacts with, first-generation isoprene nitrates are formed, namely C5 carbonyl nitrate (ICN), β and δ-hydroxy nitrate (IHNB and IHND) and β and δ-nitrooxyhydroperoxide (INPB and INPD).

We set the ICN yield at 54% and 72% for the $INO_2+NO_3$ and $INO_2+INO_2$ reactions, respectively. ICN is a first-generation isoprene nitrate that is reactive towards $NO_3$ (Rollins et al., 2009). Subsequent oxidation of ICN by $NO_3$ forms second-

generation isoprene nitrates as well as nitric acid at rates and yields based on Master Chemical Mechanism (MCM v3.2) (Jenkin et al., 1997; Saunders et al., 2003). MCM v3.2 uses a reaction rate coefficient of $1.22 \times 10^{-14}$ $cm^3$ $molec^{-1}$ $s^{-1}$ at 298 K, which is 5 times slower than the rate given by Rollins et al. (2009) and an order of magnitude faster than the rate given by Schwantes et al. (2015). Given the differences in the experimental data, splitting the difference by using the MCM rate seems a reasonable choice.

IHNB and IHND are also identified in the chamber experiments as products of the INO2+INO2 reaction. We follow Schwantes et al. (2015) and use rate constants and products, respectively from Lee et al. (2014) and Jacob et al. (2014), for the subsequent fate of IHNB and IHND upon reaction with OH. We note that these rate constants and products correspond to the rate constants and products for the reactions of ISOPND and ISOPNB (OH-initiated isoprene hydroxy nitrates) with OH. In Fig. 2, we only show reactions for one of the isomers.

Nitrooxy hydroperoxides (INPD and INPB) are the dominant products of the $INO_2+HO_2$ reaction with a combined 77% yield, most of which is δ–isomers (INPD). Schwantes et al. (2015) found that the total molar yield of INPD and INPB per reacted isoprene is higher than the yield found in previous studies (Ng et al., 2008; Kwan et al., 2012). In these previous studies, the nitrooxycarbonyl (ICN) was the main contributor to the family of isoprene nitrates produced by $NO_3$-initiated chemistry and nitrooxyhydroperoxides (INPD and INPB) were a minor fraction. The difference is likely caused by variation

in the fate of the nitrooxy peroxy radical ($INO_2+HO_2$ vs. $INO_2+NO_3$) under different experimental conditions. The fate of this radical ($INO_2$) in the nighttime atmosphere is still highly uncertain (Brown and Stutz, 2012; Boyd et al., 2015).

Chamber experiments by Schwantes et al. (2015) suggested that OH reacts with INPD and INPB to form second-generation isoprene nitrates and nitrooxy hydroxy epoxides (INHE), a newly identified product that undergoes a similar heterogeneous chemistry as IEPOX and can impact secondary organic aerosol (SOA) formation. We set the yield of the INHE δ-isomer

from INPD at 37% and yield of the INHE β-isomer from INPB at 78%. Those are lower than the IEPOX yield formed from ISHP. INHE isomers will be further oxidized by OH, leading to recycling of $NO_x$ or forming a later generation of organic nitrates. We use the same rates that we use for OH oxidation of IEPOX isomers (Schwantes et al., 2015; Bates et al., 2016). In this scheme, we also assume that INPD and INPB undergo hydrogen abstraction from the hydroperoxide group with the



same rate constants as those we use for hydrogen abstraction from the hydroperoxide group of ISHP as suggested by St. Clair et al. (2015).

C4 carbonyl hydroperoxy nitrates (R4NO) and C4 carbonyl hydroxynitrates (R4N) form from oxidation of the first-generation isoprene nitrates. In reaction with OH, they produce propanone nitrate, PROPNN. The subsequent fate of

PROPNN is dominated by oxidation by OH and photolysis resulting in the return of $NO_x$ to the atmosphere. Here, we include a photolysis rate for PROPNN and other carbonyl nitrates (ICN, ETHLN), MVKN and MACRN, that is faster than that used in some other recent mechanisms following the recommendations in Müller et al. (2014). These rates are 10 times larger than the rates given by Paulot et al. (2009a). We use the enhanced photolysis rate of PROPNN for carbonyl nitrate ICN formed from $NO_3$-initiated isoprene oxidation, and also include photolysis reactions for other new isoprene nitrate

species, following Schwantes et al. (2015) and MCM v3.2. The fast photolysis reactions of organic nitrates implemented in this work leads to an increase in $NO_x$ concentrations.

**2.2 Organic nitrate deposition to surfaces**

We use the resistance-based approach based on the original formulation of Wesely (1989) to calculate dry deposition velocities. Dry deposition rates of $RONO_2$ in RACM2_Berkeley follow Ito et al. (2007). In RACM2_Berkeley2, using the

same formalism, we update the dry deposition parameters (effective Henry's Law coefficients (H*) and reactivity factors (f0)) for isoprene nitrates as recommended by Nguyen et al. (2015), which results in more rapid removal than in RACM2_Berkeley (Table S3 in the Supplement). We also update the dry deposition parameters for monoterpene-derived nitrates that were previously assumed to deposit at a rate similar to the deposition rate of isoprene hydroxy nitrates (Browne et al., 2014; Ito et al., 2007). Dry deposition velocities of the nitrooxy hydroxy epoxides (INHE) are assumed to be similar to

the updated depositional loss rate of IEPOX, given by Nguyen et al. (2015).

**2.3 Hydrolysis of tertiary nitrates**

In addition to oxidation, photolysis and deposition to the surface, another possible fate of organic nitrates is uptake to the aerosol phase followed by hydrolysis. A rapid hydrolysis (Hu et al., 2011; Jacobs et al., 2014) is recognized for tertiary nitrates, while non-tertiary nitrates under atmospheric conditions are considered unreactive (Darer et al., 2011; Boyd et al.,

2015). Due to the limitations of the model representation of organic nitrate aerosol from either aqueous (Marais et al., 2016) or vapor-pressure dependent pathways (Pye et al., 2015), we represent this process for gas-phase organic nitrates by applying a time scale of 3h for tertiary nitrates based on the laboratory chamber study by Boyd et al. (2015). The fraction of tertiary ($F_{tertiary}$) vs. non-tertiary nitrates is estimated, depending upon the molecular structure of the nitrate, from MCM v3.2.

We apply $F_{tertiary}$ at 41% for β isomers of isoprene nitrates from OH oxidation while we assume that all δ isomers are non-

tertiary. Most of the nitrates formed by $NO_3$-initiated chemistry of isoprene are not tertiary nitrates. The fraction of tertiary nitrates for monoterpene-derived nitrates is also different for species formed by OH oxidation than from $NO_3$ oxidation. In





RACM2_Berkeley2, we introduce TONIH (C10 nitroxy hydroperoxide), TONIN (saturated) and UTONIN (unsaturated) monoterpene-derived nitrates from NO$_3$ oxidation, which differ from the unsaturated (UTONIT) and saturated (TONIT) monoterpene-derived nitrates from OH oxidation. Since contributions of tertiary limonene, α-pinene and β-pinene nitrates from NO$_3$ reaction are 35%, 15% and 50% (MCM v3.2), respectively, we define F$_{tertiary}$ at 35% as an average for TONIH,

UTONIN and TONIN. F$_{tertiary}$ is defined at 77% for UTONIT and TONIT. The value is average of the 62% for α-pinene nitrates and 92% for β-pinene nitrates and is equal to the 77% for limonene nitrates from OH chemistry.

Further changes in the RACM2_Berkeley2 mechanism for monoterpene nitrate chemistry consist of a revised reaction rate for API+NO$_3$. The rate constant is calculated as an average of the rates given in MCM v3.2 for α-pinene and β-pinene, as API in the mechanism indicates a 50-50 mixture of α-pinene and β-pinene. In our mechanism, following Browne et al.

(2014), first-generation monoterpene nitrates react with O$_3$ and OH and form second-generation nitrates. Here, we also add reaction of first-generation monoterpene nitrates with NO$_3$ with the rate constant K=3.15 x 10$^{-13}$ exp (-448.0/Temp), following Fisher et al. (2016). We assume the second-generation monoterpene nitrate can oxidize, photolyze and deposit identically to nitric acid (Browne et al., 2014). In summary, we have described a detailed chemical mechanism tracking individual organic nitrates in some detail through second-generation products of isoprene and monoterpene oxidation.

**3 Results and Discussion**

We evaluate our mechanism by comparison to SOAS observations in Bibb County, Alabama (32.90° N latitude, 87.25° W longitude) in summer 2013. These observations together with field campaign data from the long-term monitoring site in the SouthEastern Aerosol Research and CHaracterization (SEARCH) Network (Hansen et al., 2003) (at the same location) provide unique resources for evaluation of our model of organic nitrate chemistry. The measurements include total and

speciated organic nitrates, gas phase and aerosol organic nitrates, HO$_x$ radicals, a wide range of VOCs, and ozone (Pye et al., 2015; Romer et al., 2016; Lee et al., 2016).

**3.1 Organic nitrate concentrations**

Figure 3a shows the spatial distribution of total organic nitrates for the 24-h average of the model simulation period at the surface. The location of SOAS ground site (at the Centreville) is circled in the figure. The campaign area is in a location with

among the highest modeled organic nitrate concentrations in the region, up to 350 ppt. Figure 3b highlights that the modeled RONO$_2$ originating from biogenic VOCs dominate over the organic nitrates with anthropogenic VOC precursors over most of the domain. In the southeast, up to 80% of organic nitrates are biogenic. Biogenic nitrates are 40-50% in the northern portion.

Figure 4 compares median diurnal cycle of observed total organic nitrates from the SOAS campaign to the model simulation

during the simulation period. Total organic nitrates in both the gas and particle phase were measured by TD-LIF (Thermal Dissociation Laser-Induced Fluorescence, (Day et al., 2002)). Temporal variability in the total organic nitrates is reproduced



with little bias (r=0.8 and normalized mean bias (NMB) =32%). Although the mean of the simulated organic nitrates (0.26±0.19) slightly overestimates the observation mean (0.20±0.1), the medians are found within variability of the observations. The highest bias in median values and variability are observed after sunset to sunrise, which is likely caused by mismatch in vertical turbulent mixing in the simulated and actual boundary layers. Inclusion of hydrolysis as a possible fate

for tertiary organic nitrates results in significant improvement of the simulations compared to the observations (not shown here). Tertiary nitrates have shorter lifetime against hydrolysis under atmospheric conditions, compared to the lifetime against deposition (Fig. S1 in the Supplement) making them the most important sink of nitrates.

Diurnal cycles of measured and simulated $RONO_2$ have a sharp peak at 320-370 ppt around 10:00 and a slow decline through the rest of the day. Throughout the night, the mixing ratios were observed and modeled to remain nearly constant.

Around 10:00, when the highest total organic nitrates are observed during SOAS, the simulated OH-initiated and second-generation organic nitrate concentrations both reach their maximum (Fig. S2 in the Supplement). Second-generation nitrates do not show sharp variability over day and night because of their longer lifetime but they do slightly increase after sunrise (around 7:00). OH-initiated organic nitrates that can remain in the residual layer overnight contribute to the total organic nitrate during the morning. At sunrise, when OH and NO began to increase (Fig. S3 in the Supplement), OH-initiated

organic nitrates increase until they reach their maximum at around 10:00. In contrast, $NO_3$-initiated organic nitrates reach their peak mixing ratio before sunrise and immediately after sunrise, they decline sharply to a minimum concentration during the day (Fig. S2 in the Supplement). As the sun sets, NO drops to near zero and $NO_3$ production initiates the formation of organic nitrates. OH and $NO_3$-initiated reactions occur out of phase in the diurnal cycle resulting in the relatively flat diurnal profile for total organic nitrate throughout the night. Observations of individual molecules are predicted to have more

strongly varying diurnal cycles, consistent with observations (Xiong et al., 2015).

### 3.2 Organic nitrate composition

The composition of the simulated organic nitrates by our model during SOAS at Centreville site (CTR) is shown in Fig. 5a. Monoterpene nitrates are calculated to be one third of total organic nitrates, which is comparable to the Browne et al. (2014) calculation for boreal regions of North America. We define total isoprene-derived nitrates in WRF-Chem as the sum of

isoprene hydroxy nitrates, isoprene carbonyl nitrates, MVK and MACR nitrates, isoprene nitrooxyhydroperoxides, ethanal nitrate, propanone nitrate and multifunctional isoprene nitrates. We find the contribution of the total isoprene-derived nitrates to total organic nitrates to be 44%. This is consistent with the range of 25-50% observed from SEAC4RS airborne measurements taken onboard the NASA DC-8 in August–September 2013 over the southeastern US (Fisher et al., 2016). However, it is in contrast to other recent modeling studies over the Southeastern US by Mao et al. (2013) and Xie et al.

(2013) that suggested, respectively, more than 90% and 60% of total organic nitrates are from isoprene oxidation. This discrepancy is likely due to the simulated longer lifetime of these nitrates as well as omission of organic nitrates produced from monoterpenes and anthropogenic VOCs in those models.




The observed RONO$_2$ composition during SOAS is shown in Fig. 5b. The sum of the individual isomers of isoprene nitrates, terpene hydroxynitrates and terpene nitrooxyhydroperoxides were measured in the gas phase by Chemical Ionization Time of Flight Mass Spectrometry using CF$_3$O$^-$ reagent ion (Crounse et al., 2006; Schwantes et al., 2015; Teng et al., 2015) and ethyl and isopropyl nitrates were measured by gas chromatography–mass spectrometry (de Gouw et al., 2003). Similar to the

model results, the largest contributions to the total organic nitrates in the observations are isoprene oxidation products, which represent 22% against 44% in the model. Carbonyl isoprene nitrates including ICN, ETHLN and PROPNN as a fraction of total RONO$_2$ (8% in the model and <7% in the observations) and their concentrations (Fig. S4a and b in the Supplement) are reproduced well by the model. However, the model overestimates the fraction of RONO$_2$ that is isoprene hydroxynitrates and MVKN+MACRN (21% modeled vs. 13% observed). Isoprene hydroxynitrates from NO$_3$-inititated chemistry are a small

portion of the total simulated isoprene hydroxynitrates (~15%). The difference between the modeled and observed contribution of isoprene hydroxynitrates to total organic nitrates is thus more likely a result of differences between the modeled and observed nitrates that are products of OH-initiated chemistry. Insights from very recent studies by Teng et al. (2017) and Wennberg et al. (2018) suggests a larger F$_{tertiary}$ for OH-initiated isoprene hydroxynitrates than the value we calculated from MCM. That reflects a larger fraction of these nitrates to be subject of hydrolysis and thus it perhaps explains

part of the discrepancy between the model simulations and observations.

The largest difference between the modeled and observed contribution of isoprene nitrates to total organic nitrates is due to the modeled gas-phase multifunctional isoprene nitrates and isoprene nitrooxy hydroperoxides. Aerosol- and gas-phase second-generation multifunctional isoprene nitrates and aerosol-phase isoprene nitrooxy hydroperoxides were not individually measured during SOAS. Instead, total aerosol-phase organic nitrates were measured by TD-LIF, using an

activated charcoal denuder to remove gas-phase organic nitrates, and found to contribute around 40% of total organic nitrates at the SOAS CTR site (Fig. 5b). Ng et al. (2008) and Rollins et al. (2009) found isoprene oxidation can form 4-23% nitrate aerosol yields and showed multifunctional nitrates to be a dominant nitrate aerosol. If the isoprene nitrooxy hydroperoxides are favored to partition to aerosol this would explain the model-measurement discrepancy for the calculated contribution of multifunctional isoprene nitrates and isoprene nitrooxy hydroperoxide. They are simulated in the gas phase using

RACM2_Berkeley2 but we might interpret them as contributing to particle phase organic nitrate. That is consistent with the Lee et al. (2016) finding from observations of speciated particle organic nitrates during the SOAS campaign. They showed total particle organic nitrates have a dominant contribution from highly functionalized isoprene nitrates containing between six and eight oxygen atoms.

Nitrate aerosol yields for monoterpene oxidation reactions from different laboratory chamber experiments, field

measurements and modeling studies have been reported to be very high (up to 100%) (Russell and Allen, 2005; Fry et al., 2009, 2011, 2013; Pye et al., 2015; Boyd et al., 2015). Among monoterpene nitrates, NO$_3$-initiated nitrates (Ayres et al., 2015) and functionalized nitrates (Lee et al., 2016) have been shown to be an especially significant fraction of the total particle organic nitrate source at SOAS site. These findings imply that the remainder of the measured particle organic





nitrates can be attributed to mono- or sesquiterpene derived $RONO_2$ including $NO_3$-initiated terpene hydroxynitrates, terpene nitrooxyhydroperoxides and multifunctional terpene nitrates, which are simulated and present in the gas phase in our mechanism. If we interpret the aerosol nitrates to be these compounds, then we find a rough correspondence between the model and observations (see Fig. 5a and b).

In RACM2_Berkeley2, contribution from organic nitrates of anthropogenic origin is simulated to be 21% of total $RONO_2$ (referred as Other at Fig. 5a). That is higher than the 10% inferred from the observations for the measured anthropogenic organic nitrates at SOAS. Some caution should be taken in the interpretation of such a comparison, as the observations at SOAS do not represent the same species as the modeled ones and include only ethyl and propyl nitrates (Fig. 5b). In RACM2_Berkeley2, a wide range of the organic nitrates of anthropogenic origin ($C_1$-$C_5$ nitrates) are categorized into four

groups including monofunctional saturated, multifunctional unsaturated, multifunctional saturated, and aromatic-derived nitrates that are partitioned from the lumped precursors including alkanes, aromatics, alcohols and alkenes. However, the remaining unspeciated measured $RONO_2$ contribute 27% of total organic nitrates observed by TD-LIF. One hypothesis to explain this difference is that the rest of simulated organic nitrates of anthropogenic origin might be a portion of the unspecified measured $RONO_2$.

**3.3 Relationships between $RONO_2$, $O_x$ and $CH_2O$**

As ozone and total organic nitrates are produced in a common reaction with branches that yield one or the other, their observed and modeled correlation provides an additional constraint on our understanding of organic nitrates. The sum of $O_3$ and $NO_2$ is conserved on longer time scales than $O_3$ alone, accordingly we use $O_x=O_3+NO_2$ in this analysis. As shown in Fig. 6, during the daytime (from 8:00 to 18:00 local time) the modeled and observed correlations between $O_x$ and $RONO_2$ are

nearly identical. A linear fit to the observations yields a line with slope of $129\pm4$ ppbv($O_x$) ppbv($RONO_2$)$^{-1}$ and a fit to the model output yields $125\pm4$ ppbv($O_x$) ppbv($RONO_2$)$^{-1}$.

This slope has typically been used to estimate the approximate branching ratio of the entire VOC mixture ($\alpha_{eff}$) with an assumption that photochemical production is rapid as compared to removal processes; $\alpha_{eff}$ is inversely proportional to the $O_x$ vs. $RONO_2$ slope (Perring et al., 2013). The quantified $\alpha$ in the laboratory for BVOCs are much higher than typical $\alpha$'s for

anthropogenic VOCs (Perring et al. (2013) and references therein). Therefore, for regions like Southeastern U.S. where BVOCs dominate the VOC mixture, a much lower slope than our calculated value is expected. We conclude that the observed slope is reflecting the short lifetime of organic nitrates at SOAS.

Formaldehyde ($CH_2O$) is another co-product to $RONO_2$ and as Perring et al. (2009b) discussed, the slope of the $RONO_2/CH_2O$ correlation is related to the ratio of the production of both species, as both have similar lifetimes. We would

expect the slope could provide a constraint on the yield of isoprene nitrates, especially since in much of the domain isoprene is the dominant source of both $RONO_2$ and $CH_2O$. Figure 7 shows the correlation between observed $CH_2O$ and $RONO_2$ during SOAS. The slope of the best fit line, with an intercept allowed to differ from zero to consider the possibility of a





background $CH_2O$ that mixes in from the free troposphere, is found to be 0.116, consistent with previous estimates by Perring et al. (2009b) who observed a slope of 0.119 during INTEX-NA in 2004. The slope would imply an OH-initiated isoprene nitrate yield of 12% (Perring et al., 2009b) if we use a lifetime of 1.7 h at SOAS for $RONO_2$ as reported by Romer et al. (2016). This is nearly identical to the yield used in the mechanism described in this manuscript. The correlation of

modeled $CH_2O$ and total $RONO_2$ has a smaller slope of 0.085. The discrepancy between the slopes from the simulated and observed data can be attributed to model overestimation of $CH_2O$ (Fig. S5 in the Supplement).

### 3.4 Organic nitrate formation

Figure 8a shows the diurnal cycle of fractional $RONO_2$ production simulated using RACM2_Berkeley2 averaged over the boundary layer at the CTR site during SOAS. During the day, production of organic nitrates is dominated by reaction of

isoprene with OH. This is consistent with the OH reactivity (OHR) of individually measured compounds at SOAS which was dominated by reaction with isoprene (Kaiser et al., 2016; Romer et al., 2016). In contrast, the vast majority of $RONO_2$ production at night is monoterpene nitrates, which are formed as a result of $NO_3$ oxidation of nighttime monoterpene emissions. $NO_3$ chemistry of isoprene leading to isoprene nitrate formation is also found to be significant during the nighttime. This fraction from isoprene that accumulates in the boundary layer in late afternoons can reach 35% of the total

organic nitrates formed at night. In addition to investigating the relative importance of instantaneous organic nitrate production based on the VOC precursors, we calculate the fraction of isoprene nitrates produced from $NO_3$ chemistry to the total isoprene nitrate production (~44%, Fig. S6 in the Supplement) over day and night of the entire modeling period confirming the relative importance of this pathway for producing isoprene nitrates versus OH oxidation of isoprene. This finding is consistent with the modeling result (~ 40%) from Xie et al. (2013).

The fraction of anthropogenic VOC–derived organic nitrates to total simulated production of organic nitrates is estimated to be negligible; however their contribution to the simulated concentrations of organic nitrates is much higher and reached up to 0.25 (Fig. 8b). This is due to their relatively long lifetime (> 100 h lifetime to oxidation by $1\times10^6$ molecules $cm^{-3}$ of OH at 298 K and long lifetime to deposition (Henry's law constant of ~1 $Matm^{-1}$), Browne et al. (2014) and references therein) that causes their concentrations to increase with time in the boundary layer in comparison to short-lived first-generation biogenic

organic nitrates. Similarly, the fraction of second-generation nitrates formed from oxidation of first-generation isoprene and monoterpene nitrates (Fig. 8a) is predicted to be (~ 0.04), much less than the calculated contribution of their concentrations to total organic nitrate concentrations (~ 0.3) (Fig. 8b). In the next subsection we will discuss more about the loss processes and lifetime of these organic nitrates.

### 3.5 Organic nitrate lifetime

We define the lifetime of organic nitrates as the concentration of $RONO_2$ divided by the combined loss rate via all proposed loss mechanisms. The loss mechanisms include chemical loss processes (oxidation, photolysis, and hydrolysis of $RONO_2$)



and deposition. The nighttime lifetime of organic nitrates might be longer than the daytime value (and might be similar to or longer than the length of a single night). Because of uncertainties associated with simulation of the boundary layer height and organic nitrate concentrations at nighttime, we focus on the daytime lifetime as a guide for thinking about the organic nitrate fate. Figure 9 shows the estimated daytime lifetime of 2 h for first-generation biogenic organic nitrates and a longer

lifetime for the second-generation organic nitrates (3.2 h). Including organic nitrates from anthropogenic sources we estimate a fairly short overall lifetime of 2.6 h for total $RONO_2$. This short lifetime results in the less efficient transport of organic nitrates to the free troposphere and over large distances from sources. Using the SOAS field observations, Romer et al. (2016) suggested ~ 1.7 h for the atmospheric lifetime of $RONO_2$. They calculated the lifetime by the assumption that $RONO_2$ are near steady state in the afternoon. If we constrain our calculation to 12:00-16:00 and give an intercept of 40 ppt as Romer

et al. (2016) did, the overall estimated lifetime in the model is estimated to be 2.9 h, but using the production rates of organic nitrates instead of the loss rates (by assumption of the atmospheric steady state condition applied in Romer at al. (2016)) our result remarkably shows very good agreement with their finding (identical value, Fig. S7 in the Supplement). GEOS-Chem simulations by Fisher et al. (2016) reported a similar short lifetime by assuming a hydrolysis lifetime of 1 h lifetime for all tertiary and non-tertiary nitrates and not including the longer-lived small alkyl nitrates.

Accurate determination of the lifetime of organic nitrates is a major challenge for assessing the influence of organic nitrates on atmospheric chemistry. However, The estimated lifetime of ~ 3 h for organic nitrates found here as well as other studies over the Southeastern United States (Perring et al., 2009b; Romer et al., 2016; Fisher et al., 2016) is less than the range of $NO_x$ lifetimes (5.5–11 h) calculated by observational studies (e.g. Valin et al., 2013; Romer et al., 2016). Organic nitrates should therefore generally be categorized as short-lived $NO_x$ reservoirs, which remove $NO_x$ in a plume, but act as a source of

$NO_x$ in remote regions.

**3.6 $NO_x$ recycling efficiency**

To determine the fraction of $NO_x$ converted to $RONO_2$ and then released back to the gas phase as $NO_x$, the relative importance of different loss pathways of organic nitrates must be known. Oxidation and photolysis of organic nitrates recycle $NO_x$ but hydrolysis and deposition cause permanent removal of $NO_x$ from the atmosphere. Recent field studies

suggested that isoprene nitrates are removed quickly by dry deposition (Nguyen et al., 2015) and some have concluded that deposition is the primary sink of nitrates (Rosen et al., 2004; Horii et al., 2006; Horowitz et al., 2007), while others estimated that oxidation or ozonolysis is the dominant loss mechanism of isoprene nitrates (Shepson et al., 1996; Ito et al., 2007; Perring et al., 2009a; Browne et al., 2013). Similar uncertainty for the fate and dominant loss processes of monoterpene nitrates was found by Browne et al. (2014). Here, we update possible fates of organic nitrates in WRF-Chem from recent

findings including photolysis (Müller et al., 2014), oxidation and ozonolysis (Lee et al., 2014), deposition (Nguyen et al., 2015) and hydrolysis (Boyd et al., 2015) and then estimated the contribution of different fates to first- and second-generation isoprene and monoterpene nitrates (Fig. 10a and b). We note that our calculation represents the loss of the nitrate



functionality and does not include the fraction of loss processes of first-generation organic nitrates by oxidation and ozonolysis that retain the nitrate functionality by forming second-generation organic nitrates.

Figure 10b shows that the loss of second-generation organic nitrates is dominated by deposition (60%), causing permanent loss of $NO_x$. That is due to the assumed rapid depositional loss of second-generation monoterpene nitrates (deposited as fast

as $HNO_3$) in this study following Browne et al. (2014). Fractional contributions of photolysis (~25%) and oxidation (~15%) are not negligible and are much larger than those estimated by Browne et al. (2014), which is a consequence of using the rapid photolysis rates of PROPNN and ETHLN as second-generation isoprene nitrates (Müller et al., 2014). In contrast, the loss of first-generation nitrates occurs largely by the sum of chemical mechanisms that recycle $NO_x$ to the atmosphere: reaction with ozone (24%) and OH (11%) and photolysis (20%), with additional loss by deposition (15%) and hydrolysis

(~29%). Fisher et al. (2016) predicted much larger losses of $RONO_2$ by aerosol hydrolysis, (~60% of total nitrate losses), reflecting the short lifetime of nitrates with respect to hydrolysis applied in their study for all nitrates. However, with our representation of hydrolysis of nitrates in aqueous solutions (slower rate and only tertiary nitrates), hydrolysis is still important (the calculated loss rate for nitrates is ~ 3 to 11 pptv $h^{-1}$), accounting for one-third of the organic nitrate loss and leading to a large increase in $HNO_3$ production in the atmosphere. Assessing the impact of hydrolysis of nitrates on the

budget of nitric acid is beyond the scope of this work. We note that the relative contribution of nitrate hydrolysis in aqueous solutions differs widely for individual $RONO_2$ species from each other due to their different structures.

Including loss of $RONO_2$ from anthropogenic sources, we find that loss of overall $RONO_2$ via hydrolysis with an additional contribution from deposition become comparable to loss via other processes that return $NO_x$ to the atmosphere. Figure 11 shows the $NO_x$ recycling efficiency, defined as the ratio between the instantaneous production of $NO_x$ from loss of organic

nitrates and the instantaneous loss of $NO_x$ to production of organic nitrates. Since chemical degradation of nitric acid is much slower than deposition, the slope of 0.52 is interpreted as the fraction of the sequestered $NO_x$ that can be exported away from its emission region and released downwind through organic nitrate chemistry. Our finding is consistent with a recycling of about half of isoprene nitrates to $NO_x$ calculated by Paulot et al. (2009a) and Horowitz et al. (2007). Of this total, we calculated ~38% of $NO_x$ cycled back relatively quickly while first-generation nitrates are oxidizing and producing second-

generation nitrates.

## 4 Conclusions

The lifetimes of organic nitrates with respect to hydrolysis, oxidation, and deposition play an important role in the $NO_x$ budget and formation of $O_3$ and secondary organic aerosols. Analyses from recent field studies in the Southeastern United States found a lifetime of ~2-3 h for organic nitrates. By incorporating new findings from recent laboratory, field and

modeling studies into a gas-phase mechanism we provide a state-of-the-science representation of expanded organic nitrates in the WRF-Chem model. Using the updated model we are able to reproduce a short organic nitrate lifetime (2.6 h), similar to that observed during SOAS.





After adding hydrolysis as a possible fate of tertiary gas-phase biogenic organic nitrates in our mechanism and in combination with all other loss mechanism, we find that the lifetime of second-generation organic nitrates is longer that the lifetime of first-generation nitrates. We find dry deposition is the dominant loss process for second-generation organic nitrates and chemical mechanisms of ozonolysis, photolysis and oxidation that can recycle $NO_x$ to the atmosphere have a

more important role in loss of first-generation organic nitrates from the atmosphere. The contribution of tertiary nitrate hydrolysis to total organic nitrate removal from the atmosphere is found to be 30%. We find, therefore, that 52% of the $NO_x$ sequestered by production of organic nitrates can be cycled back to the atmosphere.

To accurately estimate organic nitrate lifetime, the production, loss and fate of these compounds must be well constrained. Evaluation of our updated mechanism using SOAS observations in summer 2013 indicates the model represents much of the

important chemistry governing organic nitrates. We show that the simulated concentrations of total organic nitrates, correlations with $CH_2O$ and ozone and contribution of individual $RONO_2$ to total organic nitrates are in fairly good agreement with observations at the SOAS CTR ground site. We find the largest difference between the modeled and observed contributions of individual organic nitrate compounds to total $RONO_2$ is for highly functionalized isoprene nitrates and monoterpene nitrates. We attribute this difference to possible high aerosol yields of these organic nitrate species, which

are represented in the gas phase in our mechanism. Future analysis for developing a complete representation of organic nitrate chemistry including an organic nitrate aerosol formation mechanism, either from aqueous-phase uptake or vapor-pressure partitioning onto pre-existing organic aerosol, in addition to the detailed gas-phase mechanism described here will benefit the model approximation.

**Data availability**

Measurements from the SOAS campaign are available at https://esrl.noaa.gov/csd/groups/csd7/measurements/2013senex/Ground/DataDownload (SOAS Science Team, 2013).

**Acknowledgements**

We gratefully acknowledge support from NSF grants AGS-1352972, AGS-1247421 and AGS-1628530, NOAA Office of Global Programs grant NA13OAR4310067 and NASA grant NNX 15AE37G. We thank Allen Goldstein and Pawel Konrad

Misztal for PTRTOFMS VOC data and William Brune for OH data. We also thank the SOAS field campaign team including Ann Marie Carlton.



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



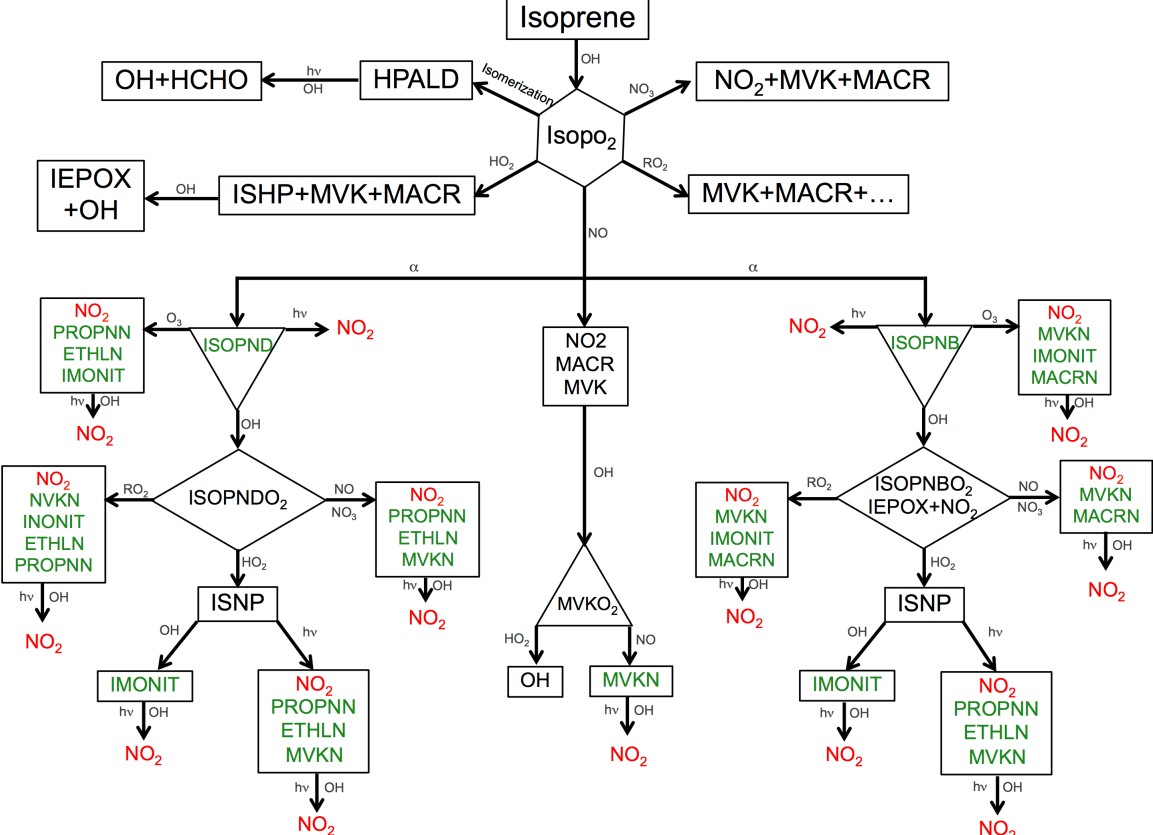

Figure 1: Schematic representation of the formation of isoprene nitrates (in green) initiated by OH oxidation. Re-released of the consumed $NO_x$ to the atmosphere by chemical loss processes of oxidation, ozonolysis and photolysis of organic nitrate is shown in red. See Table S2 in the Supplement for species descriptions.





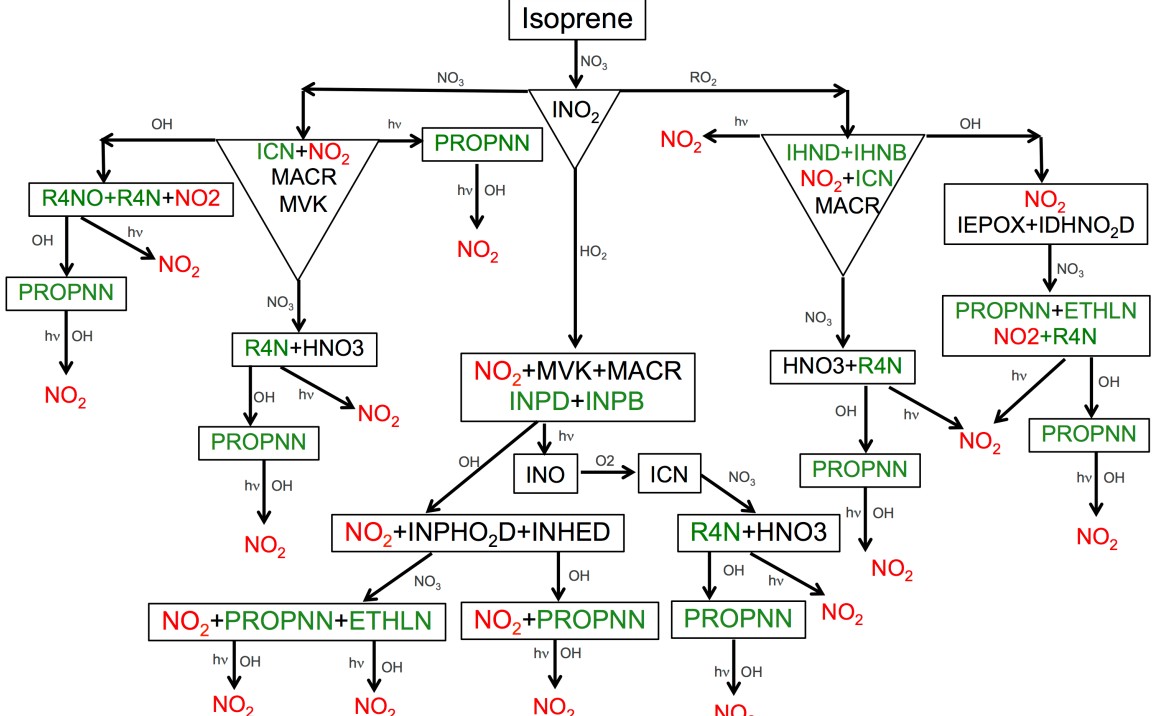

**Figure 2:** Schematic representation of the formation of isoprene nitrates (in green) initiated by $NO_3$ oxidation. For simplification, fates of only one isomer of hydroxy nitrates (IHNB and IHND) and nitrooxyhydroperoxide (INPB and INPD) are shown. See Table S2 in the Supplement for species descriptions.



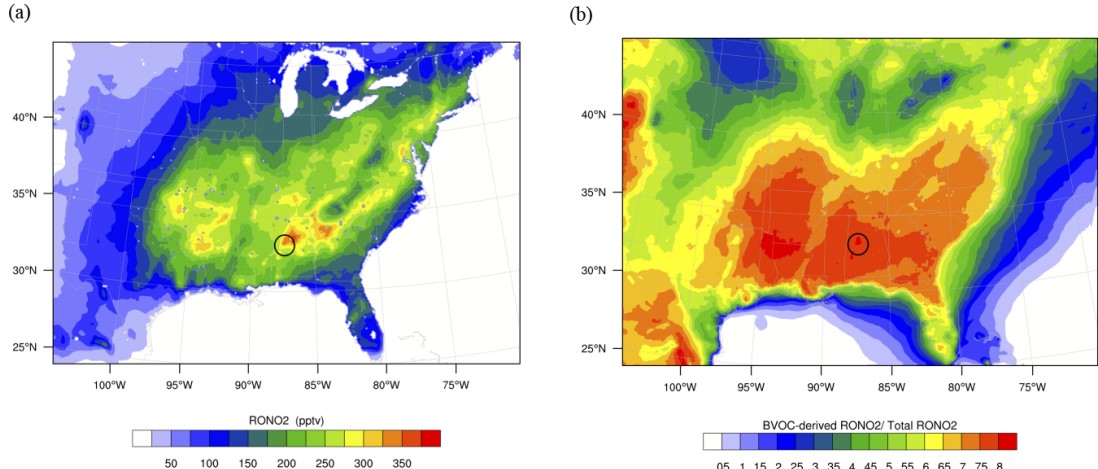

**Figure 3: (a) Concentrations of total organic nitrates for average of the model simulation period (June 2013). (b) Fractional contribution of BVOC-derived organic nitrates to total organic nitrates. The location of SOAS CTR ground site is circled in the figure.**

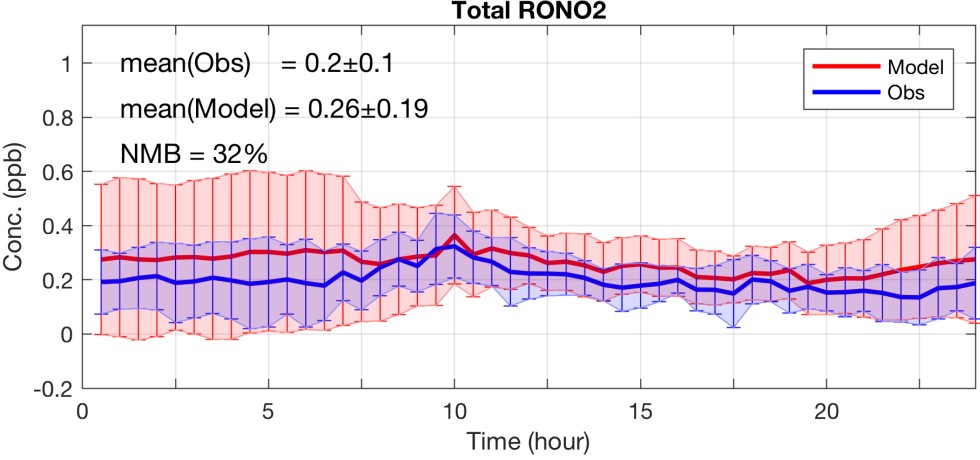

**Figure 4: Median diurnal cycles of observed and simulated total organic nitrates at Centreville during the 2013 SOAS campaign.**
10 **The vertical bars show the interquartile range of the hourly data. The panel includes mean of the simulated and observed organic nitrates and the normalized mean bias (NMB) in percent.**





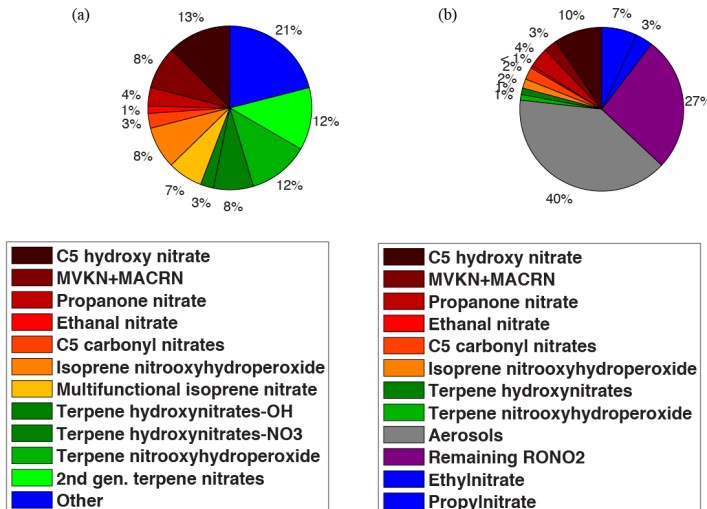

**Figure 5: The composition of the (a) simulated organic nitrates by WRF-Chem using RACM2_Berkeley2 and (b) observed organic nitrates during SOAS at CTR.**





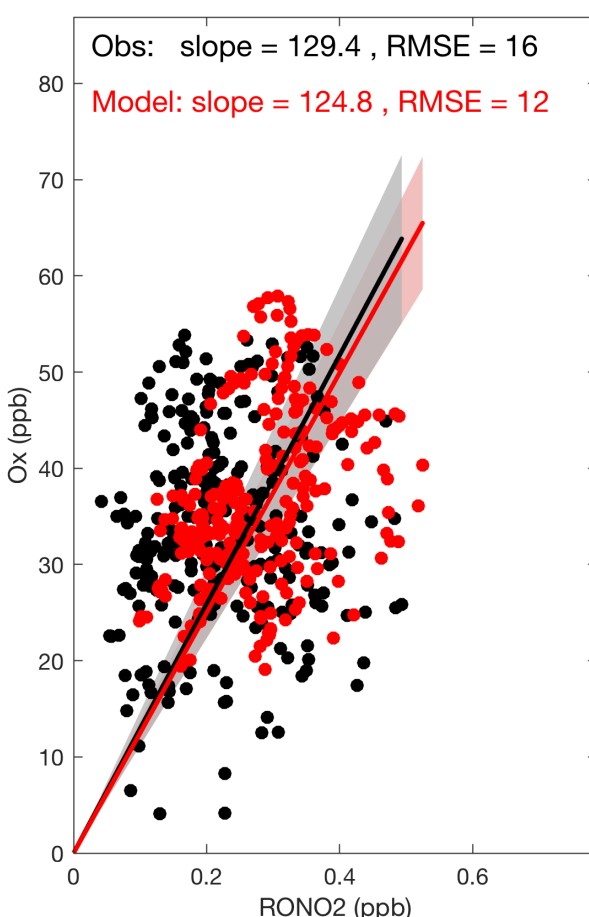

**Figure 6: The modeled and observed correlations between $O_x$ (=$O_3$+$NO_2$) and organic nitrates concentrations at daytime during SOAS. The lines indicate linear regression (intercept fixed at 0) and confidence intervals. The panel includes slopes of the lines and root mean square errors (RMSE).**





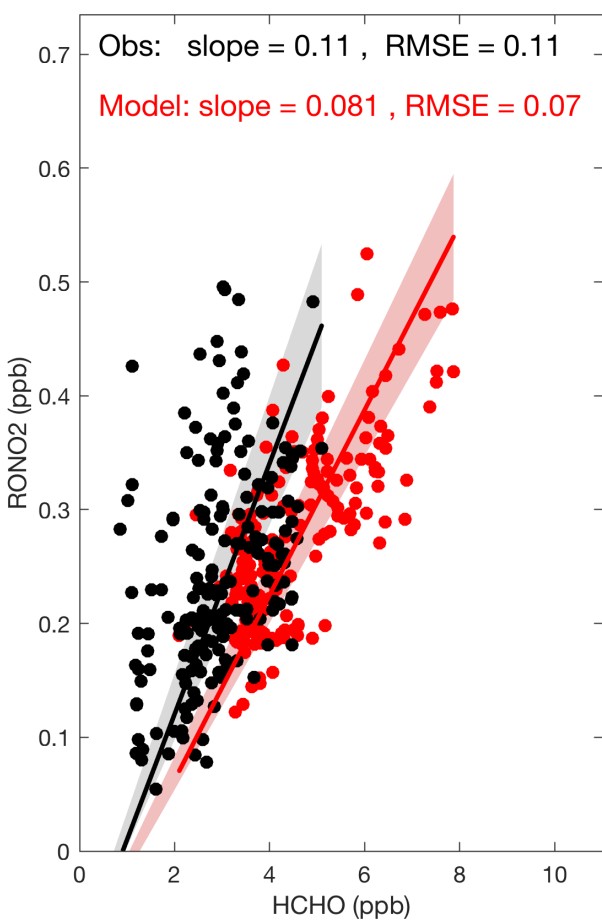

**Figure 7: The modeled and observed correlations between CH₂O and organic nitrates at daytime during SOAS. The slope shows the best fit line, with an intercept allowed to differ from zero to consider the possibility of a background CH₂O. The panel includes slopes of the lines and root mean square errors (RMSE).**





**Figure 8: Diurnal cycle of fractional organic nitrate (a) productions and (b) concentrations simulated by WRF-Chem averaged at boundary layer at the CTR site during SOAS.**



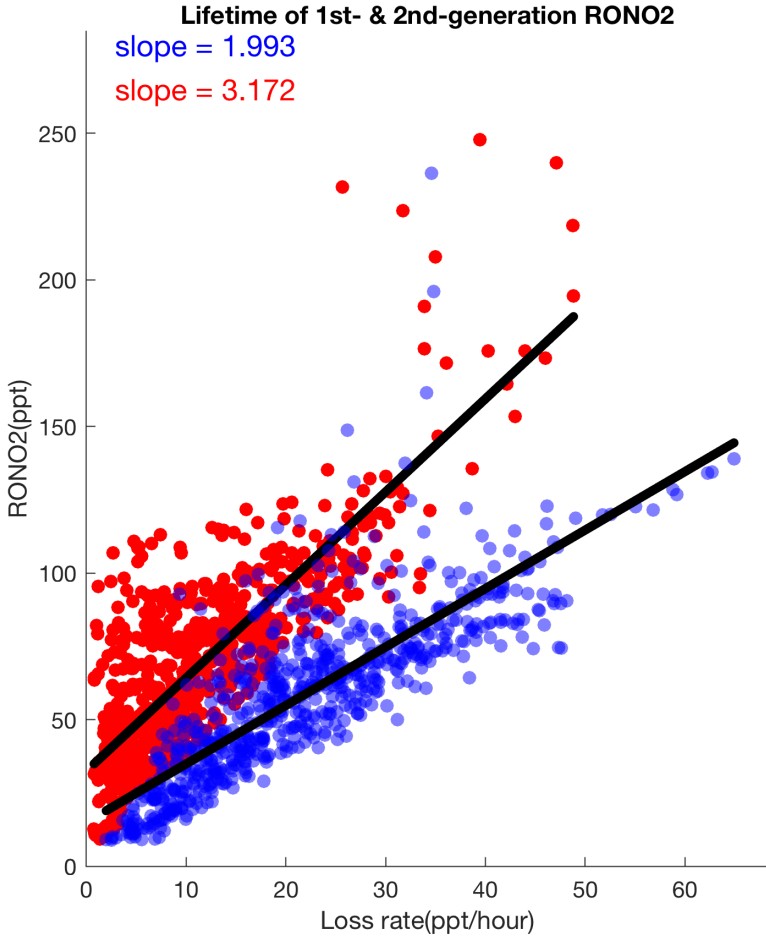

**Figure 9: The simulated concentration of 1st- (blue) and 2nd- (red) generation organic nitrates versus their loss rates at daytime during SOAS. Slopes of the linear fit give their lifetimes. The Concentrations and loss rates of 1st-genration nitrates are divided by 2.**



(a)

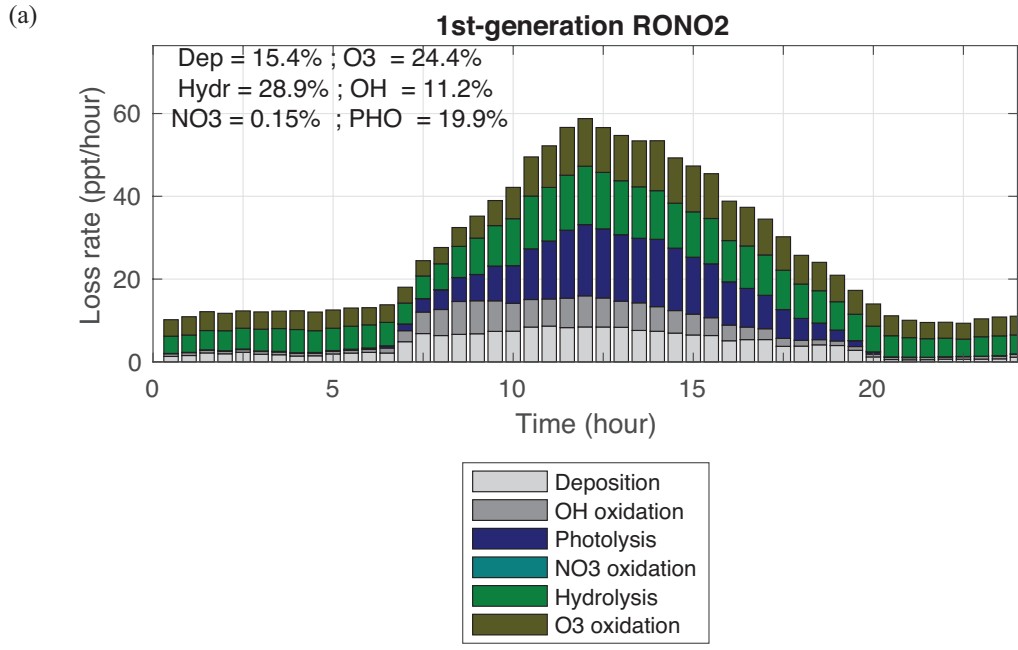

(b)

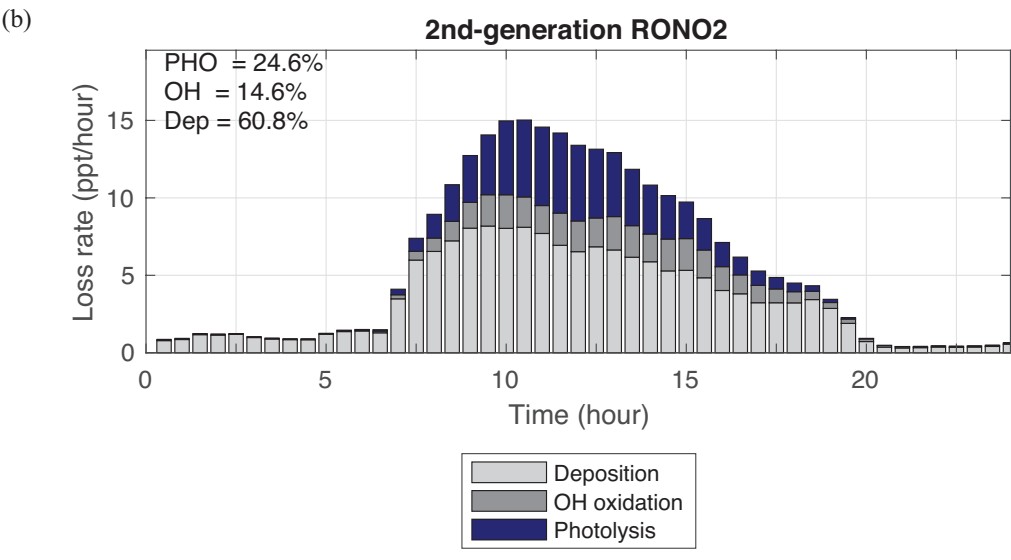

**Figure 10: Contribution of different fates to (a) the first and (b) second generation of isoprene and monoterpene nitrates loss.**



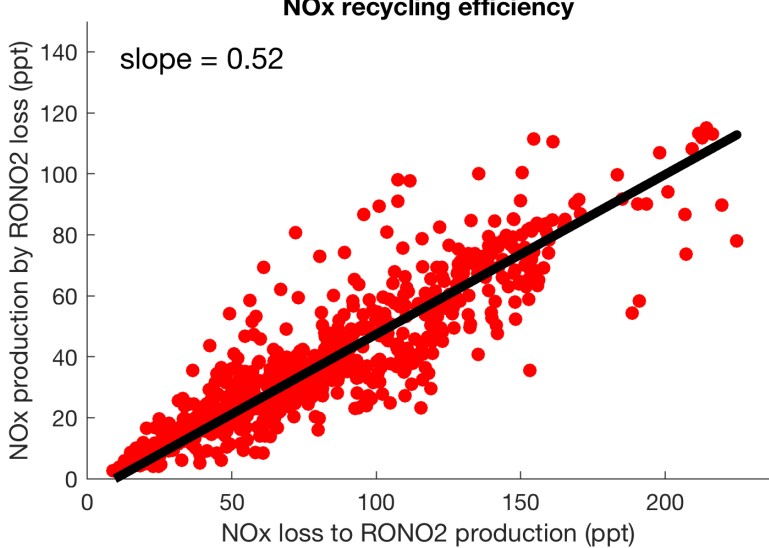

**Figure 11: The simulated instantaneous production of NOₓ from loss of organic nitrates versus the instantaneous loss of NOₓ to production of organic nitrates. The slope shows the NOₓ recycle efficiency.**

