# Peer review of "A comprehensive organic nitrate chemistry: insights into the lifetime of atmospheric organic nitrates"

_Atmospheric Chemistry and Physics, 2018_

## Referee Comment (RC1) · Anonymous Referee #1 · 9 Aug 2018

Zare et al. present a description of an updated chemical mechanism for organic nitrate chemistry, focusing on isoprene and monoterpene nitrates. They apply the mechanism to the SOAS campaign over the US Southeast to explore its agreement with observations and the implications for the lifetime of RONO2 and impacts on atmospheric NOx removal and recycling.

The paper is very well-thought out, executed, and written. It makes a nice contribution to the literature in this area. I highly recommend publication in ACP. I have only a couple minor science comments and questions for the authors to consider at their discretion. I also list separately some editorial / wording suggestions. Numbering below reflects

the page and line numbers.

Science / general comments. ======

The introduction section is very well written and provides a solid background and well-articulated motivation for the work.

9, 8-20. The discussion here misses the mark a bit. The takeaway one gets from looking at Figure 4 is how flat the entire diurnal cycle is, not just the nighttime data. So "a sharp peak at 320-370 ppt around 10:00" seems inaccurate when the whole dynamic range only spans 200-370 with quite a lot of day-to-day variability (based on the error bars). The daytime decline is obscured by the squished y-axis range of your plot. At the end of the section you give a nice description of the offsetting effects giving rise to the flatness of the data at night, but in fact these offsetting effects give rise to the flat diurnal cycle throughout the 24-h cycle, not just at nighttime. Supplemental Figure S2 shows beautifully how the flat diurnal cycle in fact represents counteracting dynamics of different nitrate species. I suggest merging Figure S2 with Figure 4 to better illustrate this point . . . for example with a separate panel, or perhaps by changing the model trace in Figure 4 to a stacked plot showing contributions from OH, NO3, and second-generation nitrates. The observed trace for the total could then be overplotted.

12, 1-6, this part is a bit confusing and can be better explained. Do you also need to assume a separate lifetime for CH2O, or is it assumed to be the same as for RONO2? If you apply the analysis to the model slope, do you arrive at the (known) actual model nitrate yield, thus confirming the applicability of the overall approach?

13, 18, "Organic nitrates should therefore generally be categorized as short-lived NOx reservoirs, which remove NOx in a plume, but act as a source of NOx in remote regions". For the purpose of the ensuing section (3.6) you state that you only consider sinks that remove the nitrate functionality, and not sinks that merely represent conversion to a different multifunctional nitrate. But it seems that is not the case for this section (3.5). Is that right? Please clarify. If that's the case, isn't the estimated ∼3h

lifetime an overly-short estimate of the degree to which the RONO2 are a short-lived NOx reservoir?

Minor technical comments. ======

2, 1: 'At modest concentrations of NOx' . . . wording is odd as it suggests that it is only at low NOx that RO2 react with NO. Perhaps "Even at modest concentrations . . . "

3, 27: please also describe the vertical resolution (e.g., number of near-surface levels, etc.).

9, 1: please clarify if r=0.8 is the correlation for the median diurnal cycle or for the whole timeseries.

11, 16, "as ozone and total organic nitrates are produced in a common reaction with branches that yield one or the other" …. It seems this is the case only for the OH-initiated nitrates, correct?

12, 24, "which causes their concentrations to increase with time in the boundary layer", not really increasing with time but rather persisting longer, leading to higher ambient concentrations for a given source, consider rewording

Figure 9, Consider secondary x and y-axis to clarify that the 1st-gen nitrates are scaled by 0.5.

Wording suggestions. ======

2, 11: missing period

3, 2: perhaps "from the atmosphere"

3, 3: "in simulations of NOx and O3" or "in simulating NOx and O3"

3, 23-25: awkward, run-on sentence

4, 2: "initial conditions"

5, 4: "reacts with OH"

5, 13, "yields of"

5, 19, awkward, perhaps "to yield either NOx or second-generation organic nitrates"

8, 24, "at Centreville"

9, 2: "observational mean", "found to be"

9, 3, "the highest bias in the model median values and variability"

9, 22, suggest "The composition of our model-simulated organic nitrates during . . ."

10, 14, suggest "that suggests a larger fraction of these nitrates is subject to . . ."

10, 21, "isoprene oxidation by NO3"

11, 5, "the contribution from"

11, 6, "from the observations of the measured"

11, 12, "contributes 27% of the total"

11, 13, "the rest of the simulated"

11, 32, "of background CH2O"

13, 6, "results in less efficient"

13, 31, "and then estimate"

14, 16, suggest deleting "from each other"

Fig 1 caption, "Re-release"

Figure 3 caption, "for the average"

Figure 4 caption, "includes the mean"

Figure 6, 7, and 9 captions, "during daytime at SOAS" rather than "at daytime during

SOAS"

Figure 7 caption, "of background"

Figure 8 caption, "production" and "averaged over the boundary layer"

Figure 9, "Concentrations" should not be capitalized.

Figure 11 caption, "recycling efficiency"

---

## Referee Comment (RC2) · Anonymous Referee #2 · 19 Aug 2018

This is a nice paper that looks into details of organic nitrate chemistry, with recent new understanding on this topic. The authors develop a new mechanism in WRF-Chem model and compare model simulations to observations in Southeast US during SOAS 2013. They find that their model is generally in good agreement with observations, assuming organic nitrates is short lived with a lifetime of 2-3h. The paper is well written. I would recommend publication on ACP after the following comments are addressed:

1. As organic nitrates are largely driven by biogenic VOCs, it is important for authors to evaluate isoprene and monoterpene concentrations in their model. Isoprene and monoterpene measurements have been shown in Fisher et al. [2016]. I assume that

they are available for comparison.

2. The authors have done a detailed comparison with Fisher et al. [2016]. It is important to point out that Fisher et al. [2016] assumes a 9% yield for first generation isoprene nitrates, while it is assumed 11.7% in this paper. Given the higher yield and slower aerosol hydrolysis in this study, could authors comment on why these two studies show similar amount of total organic nitrates in their models?

3. The authors appear to have ignored another model study on this topic, Li et al. [2018]. It seems that Li et al. [2018] also did a detailed analysis on first- and second-generation isoprene nitrates using data collected in Southeast US. It might be worthwhile to compare this model to their results in details.

4. I would suggest that the authors include two review papers on this topic in the Introduction part, Carlton et al. [2018] and Mao et al. [2018].

5. It might be useful to mention vertical resolution of WRF-Chem, to help reader understand how well the model is representing nighttime boundary layer emission and chemistry.

6. Page 10, Line 25, "They showed total particle organic nitrates have a dominant contribution from highly functionalized isoprene nitrates containing between six and eight oxygen atoms." Is this correct about the isoprene nitrates dominating particle organic nitrates? If not, then this should not be the reason for "the difference between the modeled and observed contribution of isoprene nitrates to total organic nitrates".

Reference

Carlton, A. G., et al. (2018), Synthesis of the Southeast Atmosphere Studies: Investigating Fundamental Atmospheric Chemistry Questions, Bull. Amer. Meteorol. Soc., 99(3), 547-567, doi:10.1175/bams-d-16-0048.1.

Fisher, J. A., et al. (2016), Organic nitrate chemistry and its implications for nitrogen budgets in an isoprene- and monoterpene-rich atmosphere: constraints from aircraft (SEAC4RS) and ground-based (SOAS) observations in the Southeast US, Atmos. Chem. Phys., 16(9), 5969-5991, doi:10.5194/acp-16-5969-2016.

Li, J., et al. (2018), Decadal changes in summertime reactive oxidized nitrogen and surface ozone over the Southeast United States, Atmos. Chem. Phys., 18(3), 2341-2361, doi:10.5194/acp-18-2341-2018.

Mao, J., et al. (2018), Southeast Atmosphere Studies: learning from model-observation syntheses, Atmos. Chem. Phys., 18(4), 2615-2651, doi:10.5194/acp-18-2615-2018.

---

## Author Comment (AC1) · 22 Sep 2018

We thank two reviewers for their positive and constructive comments. Our responses to the comments are provided below. The reviewers' comments are in bold, our responses in normal text, and changes made to the manuscript are shown in red italics block quotes. Page and line numbers refer to the first submission.

**Response to Referee #1**

**Zare et al. present a description of an updated chemical mechanism for organic nitrate chemistry, focusing on isoprene and monoterpene nitrates. They apply the mechanism to the SOAS campaign over the US Southeast to explore its agreement with observations and the implications for the lifetime of RONO2 and impacts on atmospheric NOx removal and recycling. The paper is very well-thought out, executed, and written. It makes a nice contribution to the literature in this area. I highly recommend publication in ACP. I have only a couple minor science comments and questions for the authors to consider at their discretion. I also list separately some editorial / wording suggestions. Numbering below reflects the page and line numbers.**

**Science / general comments. ======**
**The introduction section is very well written and provides a solid background and well-articulated motivation for the work.**

**9, 8-20. The discussion here misses the mark a bit. The takeaway one gets from looking at Figure 4 is how flat the entire diurnal cycle is, not just the nighttime data. So "a sharp peak at 320-370 ppt around 10:00" seems inaccurate when the whole dynamic range only spans 200-370 with quite a lot of day-to-day variability (based on the error bars). The daytime decline is obscured by the squished y-axis range of your plot. At the end of the section you give a nice description of the offsetting effects giving rise to the flatness of the data at night, but in fact these offsetting effects give rise to the flat diurnal cycle throughout the 24-h cycle, not just at nighttime. Supplemental Figure S2 shows beautifully how the flat diurnal cycle in fact represents counteracting dynamics of different nitrate species. I suggest merging Figure S2 with Figure 4 to better illustrate this point . . . for example with a separate panel, or perhaps by changing the model trace in Figure 4 to a stacked plot showing contributions from OH, NO3, and second-generation nitrates. The observed trace for the total could then be overplotted.**

We have merged Figures 4 and S2 and have shown contributions from OH-, NO3-initiated and second-generation organic nitrates as stacked bars in the figure. We have also reduced y-axis range to show the diurnal variability more clearly, so that maximum values around 10:00 and slow decline through the rest of the day have become more recognizable. However, we agree with the reviewer, the modeled organic nitrates at nighttime show more uncertain and higher values than observations which make the peak harder to distinguish in simulated diurnal variability. That can be due to mismatch in vertical turbulent mixing in the simulated and actual boundary layers. We have clarified this matter in the text (page 9, line 4).

We have revised the text as follows:

*Page 9, 8-9 "Diurnal cycles of measured and simulated RONO$_2$ have  maximum values at 320 and 370 ppt around 10:00, respectively, with a slow decline through the rest of the day."*

[Figure]

"*Figure 4: Median diurnal cycles of observed (black) and simulated (red) total organic nitrates at Centreville during the 2013 SOAS campaign. The vertical bars show the interquartile range of the hourly data. The panel includes the mean of the simulated and observed organic nitrates. Diurnal cycle of the OH-initiated, $NO_3$-initiated and second-generation organic nitrate concentrations are shown as the stacked bars.*"

**12, 1-6, this part is a bit confusing and can be better explained. Do you also need to assume a separate lifetime for CH2O, or is it assumed to be the same as for RONO2? If you apply the analysis to the model slope, do you arrive at the (known) actual model nitrate yield, thus confirming the applicability of the overall approach?**

We have added a reference to Perring et al.,that estimated the short lifetime $CH_2O$, in the text and described the assumptions more clearly

*Page 11, 28-29 "Formaldehyde ($CH_2O$) is another co-product to $RONO_2$ and as Perring et al. (2009b) discussed, the slope of the $RONO_2$ /$CH_2O$ correlation is related to the ratio of the production of both species, as both have similar lifetimes (Perring et al., 2009b).*

Using the simulated $RONO_2$ and $CH_2O$ we do not exactly derive the isoprene nitrates yield used in the model. The text (page 12) discussing the issue is as follows:

*Page 12, 2-6 "The slope would imply an OH-initiated isoprene nitrate yield of 12% (Perring et al., 2009b) if we use a lifetime of 1.7 h at SOAS for $RONO_2$ as reported by Romer et al. (2016). This is nearly identical to the yield used in the mechanism described in this manuscript. However, the correlation of modeled $CH_2O$ and modeled total $RONO_2$ has a smaller slope of 0.085. The discrepancy between the slopes from the simulated and observed data can be attributed to model overestimation of $CH_2O$ (Fig. S5 in the Supplement).*

**13, 18, "Organic nitrates should therefore generally be categorized as short-lived NOx reservoirs, which remove NOx in a plume, but act as a source of NOx in remote regions". For the purpose of the ensuing section (3.6) you state that you only consider sinks that remove the nitrate functionality, and not sinks that merely represent conversion to a different multifunctional nitrate. But it seems that is not the case for this section (3.5). Is that right? Please clarify. If that's the case, isn't the estimated ~3h lifetime an overly-short estimate of the degree to which the RONO$_2$ are a short-lived NOx reservoir?**

We derive an ~3hr lifetime of the nitrate functional group to conversion to NO$_x$ or HNO$_3$. Some of the individual first and second generation molecules have longer lifetimes. We have added some detail in the Supplement as follows to be clearer about our thinking:

*"Additional model documentation*
*Equations*

*To compute the NOx recycling efficiency (NRE) and RONO$_2$ lifetime ($\tau_{RONO2}$) we use Eq (1) and Eq (2):*

$$NRE = \frac{P(NOx)}{Loss(NOx)} \quad (1)$$

$$\tau_{RONO2} = \frac{[RONO2]}{Loss(RONO2)} \quad (2)$$

*where P (NOx) and Loss (NOx) refer to the re-released NOx due to oxidation and photolysis of RONO$_2$, and loss of NO$_x$ due to the production of RONO$_2$, respectively. Loss (RONO$_2$) is loss rate of RONO$_2$. This lifetime does not include reactions that convert one nitrate into a different nitrate. In contrast, to calculate the lifetime of specific individual molecules we consider all reactions.*

*A simplified scheme, as an example, provides more detail on the approach used.*

| Reactants | Products | species to track rates |
|---|---|---|
| BVOC + OH | RO2 | |
| RO2 + NO | α AN1 + (1- α) NO2 | + α LNOX |
| AN1 + OH/O3/hv | γ AN2 + (1- γ) NO2 | + (1- γ) PNOX1 +LAN1 |
| AN2 + OH/hv | NO2 | + PNOX2 + LAN2 |

*LAN1, LAN2, LNOX are used to track insatantanous loss of first- and second-generation RONO2 (AN1 and AN2) and NOx at each time step. PNOX1and PNOX2 track instantaneous re-released NOx due to loss of first- and second-generation RONO2. Thus, NOx recycling efficiency and lifetime of first- and total RONO2 at each time step are calculated as:*

$$NRE = \frac{((1-\gamma)\,PNOX1 + PNOX2)}{(\alpha\,LNOX)}$$

$$\tau_{AN1} = \frac{[AN1]}{(LAN1)}$$

$$\tau_{RONO2} = \frac{[AN1] + [AN2]}{((1 - \gamma)PNOX1 + LAN2)}$$

"

**Minor technical comments. ======**

**2, 1: 'At modest concentrations of NOx' . . . wording is odd as it suggests that it is only at low NOx that RO2 react with NO. Perhaps "Even at modest concentrations . . . "**

We have modified the sentence as follows:

*Page 2, 1-2 "At high and even modest concentrations of NO$_x$, the peroxy radicals react primarily with NO"*

**3, 27: please also describe the vertical resolution (e.g., number of near-surface levels, etc.).**

This comment is in common with Reviewer 2's comment. We have expanded the text to include more information about the vertical coordinate.

*Page 3, 26-30 "We use WRF-Chem version 3.5.1 (Grell et al., 2005) with a horizontal resolution of 12 km  over the eastern United States. Our simulation domain is defined on the Lambert projection, which is centered at 35°N, 87°W and has 290 and 200 grid points in the west–east and south–north directions, respectively (see Fig. 3 for the horizontal domain). The vertical coordinate is hybrid sigma-pressure that covers 30 levels from the surface to 100 hPa. Near surface levels follow terrain and gradually transitions to constant pressure at higher levels. Vertical grid spacing varies with height such that finer spacing is assigned to the lower atmosphere while coarser vertical spacing is applied at higher levels. In this analysis, the model predictions are averaged over two lowest model levels (~25 m) used for comparison with ground-based measurements taken from a 20 m walk-up tower. The predicted concentrations in boundary layer are described as an average over 8 vertical model levels with a height (~1000 m) that is comparable with the planetary boundary layer depth at midday at Southeastern United States in June 2013."*

**9, 1: please clarify if r=0.8 is the correlation for the median diurnal cycle or for the whole timeseries.**

This r$^2$ shows correlation for the whole time series. We have clarified this as:

*Page 9, 1 "Temporal variability in the total organic nitrates for the entire time series is reproduced with little bias (r$^2$=0.8 and normalized mean bias (NMB) =32%)."*

**11, 16, "as ozone and total organic nitrates are produced in a common reaction with branches that yield one or the other" . . .. It seems this is the case only for the OH initiated nitrates, correct?**

Correct, We have modified the sentence as follows:

*Page 11, 16-17 " During daytime,  ozone and  organic nitrates are produced in a common reaction with branches that yield one or the other. Therefore, their observed and modeled correlation provides an additional constraint on our understanding of organic nitrates."*

**12, 24, "which causes their concentrations to increase with time in the boundary layer", not really increasing with time but rather persisting longer, leading to higher ambient concentrations for a given source, consider rewording**

We have changed the text as follows:

*Page 12, 23-25 "This is due to their relatively long lifetime (> 100 h lifetime to oxidation by $1 \text{Å} \sim 10^6$ molecules $cm_{-3}$ of OH at 298 K and a similarly long lifetime to deposition (Henry's law constant of $\sim 1 Matm_{-1}$ ), Browne et al. (2014) and references therein) that causes ~~their concentrations to increase with time in the boundary layer in comparison to short-lived first-generation biogenic organic nitrates them to persist longer in the atmosphere."*

**Figure 9, Consider secondary x and y-axis to clarify that the 1st-gen nitrates are scaled by 0.5.**

Done. The figure is revised accordingly.

[Figure]

*"Figure 9: The simulated concentration of 1st- (blue) and 2nd- (red) generation organic nitrates versus their loss rates during daytime at SOAS. Slopes of the linear fit give their lifetimes. The concentrations and loss rates of 1st-genration nitrates are divided by 2."*

**Wording suggestions. ======**
**2, 11: missing period**
**3, 2: perhaps "from the atmosphere"**
**3, 3: "in simulations of NOx and O3" or "in simulating NOx and O3"**
**3, 23-25: awkward, run-on sentence 4, 2: "initial conditions"**
**5, 4: "reacts with OH"**
**5, 13, "yields of"**
**5, 19, awkward, perhaps "to yield either NOx or second-generation organic nitrates"**
**8, 24, "at Centreville"**
**9, 2: "observational mean", "found to be"**
**9, 3, "the highest bias in the model median values and variability"**
**9, 22, suggest "The composition of our model-simulated organic nitrates during . . ."**
**10, 14, suggest "that suggests a larger fraction of these nitrates is subject to . . ."**
**10, 21, "isoprene oxidation by NO3"**
**11, 5, "the contribution from"**
**11, 6, "from the observations of the measured"**
**11, 12, "contributes 27% of the total"**
**11, 13, "the rest of the simulated"**
**11, 32, "of background CH2O"**
**13, 6, "results in less efficient"**
**13, 31, "and then estimate"**
**14, 16, suggest deleting "from each other"**
**Fig 1 caption, "Re-release"**
**Figure 3 caption, "for the average"**
**Figure 4 caption, "includes the mean"**
**Figure 6, 7, and 9 captions, "during daytime at SOAS" rather than "at daytime during SOAS"**
**Figure 7 caption, "of background"**
**Figure 8 caption, "production" and "averaged over the boundary layer"**
**Figure 9, "Concentrations" should not be capitalized.**
**Figure 11 caption, "recycling efficiency"**

All wording suggestions are applied to the text.

---

## Author Comment (AC2) · 22 Sep 2018

We thank two reviewers for their positive and constructive comments. Our responses to the comments are provided below. The reviewers' comments are in bold, our responses in normal text, and changes made to the manuscript are shown in red italics block quotes. Page and line numbers refer to the first submission.

**Response to Referee #2**

**This is a nice paper that looks into details of organic nitrate chemistry, with recent new understanding on this topic. The authors develop a new mechanism in WRF-Chem model and compare model simulations to observations in Southeast US during SOAS 2013. They find that their model is generally in good agreement with observations, assuming organic nitrates is short lived with a lifetime of 2-3h. The paper is well written. I would recommend publication on ACP after the following comments are addressed:**

**1. As organic nitrates are largely driven by biogenic VOCs, it is important for authors to evaluate isoprene and monoterpene concentrations in their model. Isoprene and monoterpene measurements have been shown in Fisher et al. [2016]. I assume that they are available for comparison.**

We have added the requested figure to the Supplement and described it in the text as:

*Page 12, 6 "In Fig S5, we also provide additional model evaluation for isoprene and monoterpene concentrations."*

[Figure]

*"Figure S4: Median diurnal cycles of observed and simulated CH₂O, isoprene and monoterpenes at Centreville during the 2013 SOAS campaign. The vertical bars show the interquartile range of the hourly data. The panel includes mean of the simulated and observed values."*

**2. The authors have done a detailed comparison with Fisher et al. [2016]. It is important to point out that Fisher et al. [2016] assumes a 9% yield for first generation isoprene nitrates, while it is assumed 11.7% in this paper. Given the higher yield and slower aerosol hydrolysis in this study, could authors comment on why these two studies show similar amount of total organic nitrates in their models?**

Our predicted RONO2 concentrations are within the observed variability but we have estimated the mean of total RONO2 (~260 ppt) to be higher than the value (~200 ppt) reported in Fisher et al. (2016). This difference is not too dissimilar from the difference in yields. In addition, the higher yield in the mechanism in our paper is balanced by more rapid deposition of second-generation monoterpene nitrates (following Browne et al., 2014) than in Fisher et al. (2016).

**3. The authors appear to have ignored another model study on this topic, Li et al. [2018]. It seems that Li et al. [2018] also did a detailed analysis on first- and second generation isoprene nitrates using data collected in Southeast US. It might be worthwhile to compare this model to their results in details.**

Thanks to the reviewer for pointing out this oversight. Key differences are summarized in the following table and we have added a discussion comparing our results to Li et al. (2018) to the paper as follows.

|  | This study | Li et al. (2018) |
|---|---|---|
| Model | Chemical transport WRF-Chem v3.5 model | AM3 global chemistry–climate model |
| Horizontal Resolution | 12 km | 50 km |
| Isoprene nitrate yield | 11.7% (yield of β vs. δ isomers are 10.5% and 1.2% respectively) | 10% (only β isomer) |
| Isoprene NO3 chemistry | Following Schwantes et al. (2015) | Based on the Leeds Master Chemical Mechanism (MCM v3.2) |
| Monoterpenes nitrate yield from OH chemistry | 18% for low-reactivity monoterpenes and 22% for high-reactivity monoterpenes (following Browne et al., 2014) | Simplified monoterpenes nitrate chemistry with an organic nitrate yield 26% for one lumped monoterpenes |
| Monoterpenes nitrate yield from NO3 chemistry | 10% for low-reactivity monoterpenes and 70% for high-reactivity monoterpenes (following Browne et al., 2014) | 10% for one lumped monoterpenes |
| Hydrolysis of RONO2 | hydrolysis of gas-phase tertiary | 2-step hydrolysis scheme: |

| | organic nitrates (hydrolysis lifetime = 3 hr) | heterogeneous uptake of organic nitrates onto aerosols and then hydrolysis of aerosol-phase nitrates (hydrolysis lifetime = 3 hr). In base case, only ISOPNB is assumed to hydrolyze. |
| --- | --- | --- |

*Page 9, 1-3 "Temporal variability in the total organic nitrates is reproduced with little bias ($r^2$=0.8 and normalized mean bias (NMB) =32%). Although the mean of the simulated organic nitrates (0.26± 0.19) slightly overestimates the mean of the observations (0.20± 0.1), the medians are within the variability of the observations. The simulated mean of total $RONO_2$ in this study is in the range of two other recent modeling studies over the Southeastern US in summer 2013 that simulated 200 ppt (Fisher et al., 2016) and 270 ppt (Li et al, 2018). However, in both of these studies $RONO_2$ derived from anthropogenic VOC precursors were not included. In our simulation, these organic nitrates represent ~20% of total $RONO_2$. Specific sources of the differences include, the slightly smaller yield of 10% yield for isoprene nitrates and application of a 3 hr hydrolysis lifetime only for ISOPNB in Li et al., (2018). Fisher et al. (2016) apply a faster hydrolysis rate (1hr) for all organic nitrates and a lower yield (9% for isoprene nitrates)."*

*Page 9, 4-8 "Inclusion of hydrolysis as a possible fate for tertiary organic nitrates results in significant improvement of the simulations compared to the observations (not shown here). Tertiary nitrates have shorter lifetime against hydrolysis under atmospheric conditions, compared to the lifetime against deposition (Fig. S1 in the Supplement) making them the most important sink of nitrates. Li et al. (2018) also showed, by introducing the hydrolysis of ISOPNB, the model relative bias of total $RONO_2$ was reduced 18% during ICARTT (summer 2004) over the Southeastern United States."*

*Page 10, 31 "Among monoterpene nitrates, $NO_3$ -initiated nitrates (Ayres et al., 2015) and functionalized nitrates (Lee et al., 2016) have been shown to be an especially significant fraction of the total particle organic nitrate source at SOAS site. These findings imply that the remainder of the measured particle organic nitrates can be attributed to mono- or sesquiterpene derived $RONO_2$ including $NO_3$ -initiated terpene hydroxynitrates, terpene nitrooxyhydroperoxides and multifunctional terpene nitrates, which are simulated and present in the gas phase in our mechanism. If we interpret the aerosol nitrates to be these compounds, then we find a rough correspondence between the model and observations (see Fig. 5a and b). However, Li et al. (2018) estimated a smaller contribution of gas-phase $NO_3$ -initiated monoterpene nitrates to total $RONO_2$ due to a lower molar yield (10% vs 70% for high-reactivity monoterpenes and 10% for low-reactivity monoterpenes in this study). In contrast, due to other differences in the mechanism they found a larger contribution of OH-initiated monoterpene nitrates to total $RONO_2$ than our finding in this study."*

*Page 13,13-15 "GEOS-Chem simulations by Fisher et al. (2016) reported a similar short lifetime by assuming a hydrolysis lifetime of 1 h lifetime for all tertiary and non-tertiary nitrates and not including the longer-lived small alkyl nitrates. However, Li et al., (2018) estimated longer lifetimes for individual nitrates except ISOPNB, which they assumed to be hydrolyzed.*

**4. I would suggest that the authors include two review papers on this topic in the Introduction part, Carlton et al. [2018] and Mao et al. [2018].**

We add these references to the introduction and result sections.

*Page 1, 26-29 " The oxidative chemistry of BVOCs affects the distribution of oxidants (OH, O₃ , NO₃ ) and the lifetime of NOₓ (=NO+NO₂ ), creating a feedback loop that affects oxidant concentrations, the lifetime of BVOCs and secondary organic aerosol formation (Carlton et al., 2018; Mao et al., 2018)."*

*Page 8, 16-18 "We evaluate our mechanism by comparison to SOAS observations in Bibb County, Alabama (32.90° N latitude, 87.25° W longitude) in summer 2013 (Carlton et al., 2018; Mao et al., 2018)."*

**5. It might be useful to mention vertical resolution of WRF-Chem, to help reader understand how well the model is representing nighttime boundary layer emission and chemistry.**

This comment was in common with Reviewer 1's comment. We have added this information at the text as:

*Page 3, 26-30 "We use WRF-Chem version 3.5.1 (Grell et al., 2005) with a horizontal resolution of 12 km  over the eastern United States. Our simulation domain is defined on the Lambert projection, which is centered at 35°N, 87°W and has 290 and 200 grid points in the west–east and south–north directions, respectively (see Fig. 3 for the horizontal domain). The vertical coordinate is hybrid sigma-pressure that covers 30 levels from the surface to 100 hPa. Near surface levels follow terrain and gradually transitions to constant pressure at higher levels. Vertical grid spacing varies with height such that finer spacing is assigned to the lower atmosphere while coarser vertical spacing is applied at higher levels. In this analysis, the model predictions are averaged over two lowest model levels used for comparison with ground-based measurements taken from a 20 m walk-up tower. The predicted concentrations in boundary layer are described as an average over 8 vertical model levels with a height that is comparable with the planetary boundary layer depth at midday at Southeastern United States in June 2013."*

**6. Page 10, Line 25, "They showed total particle organic nitrates have a dominant contribution from highly functionalized isoprene nitrates containing between six and eight oxygen atoms." Is this correct about the isoprene nitrates dominating particle organic nitrates? If not, then this should not be the reason for "the difference between the modeled and observed contribution of isoprene nitrates to total organic nitrates".**

Lee et al., (2016) have shown that "Each carbon number group in the particle phase exhibits a bell-shaped distribution, with the dominant contribution from ON typically comprising between six and eight oxygen atoms". And we have found that "The largest difference between the modeled and observed contribution of isoprene nitrates to total organic nitrates is due to the modeled gas-phase multifunctional isoprene nitrates and isoprene nitrooxy hydroperoxides." Accordingly, we have concluded that part of modeled gas-phase multifunctional isoprene nitrates can correspond with the part of observed particle organic nitrates. We have revised text as follows:

*Page 10, 24-28 "They are simulated in the gas phase using RACM2_Berkeley2 but we might interpret them as contributing to particle phase organic nitrate. That is consistent with the Lee et al. (2016) finding from observations of speciated particle organic nitrates during the SOAS campaign. They showed  particle  isoprene nitrates have a dominant contribution from highly functionalized isoprene nitrates containing between six and eight oxygen atoms."*